

# A vertebra of a small species of *Pachycetus* from the North Sea and its inner structure and vascularity compared with other basilosaurid vertebrae from the same site

Henk Jan van Vliet[1], Mark E.J. Bosselaers[2,3], Dirk K. Munsterman[4], Marcel L. Dijkshoorn[5], Jeffrey Joël de Groen[5] and Klaas Post[6]

[1] Altrecht, Utrecht, the Netherlands
[2] Koninklijk Belgisch Instituut voor Natuurwetenschappen, Brussel, Belgium
[3] Koninklijk Zeeuwsch Genootschap der Wetenschappen, Middelburg, the Netherlands
[4] Geological Survey of the Netherlands, Netherlands Organisation for Applied Scientific Research TNO, Utrecht, the Netherlands
[5] Erasmus Universitair Medisch Centrum Rotterdam, Rotterdam, the Netherlands
[6] Het Natuurhistorisch (Natural History Museum, Rotterdam), Rotterdam, the Netherlands

Corresponding author
Henk Jan van Vliet,
henkjanvanvliet@yahoo.com

## ABSTRACT

In the Western Scheldt Estuary near the Belgian-Dutch border, middle to late Eocene strata crop out at the current seafloor. Most vertebrae of large Eocene basilosaurid taxa from this area were previously described in several papers. They represent three morphotypes: elongated vertebrae of a large species of *Pachycetus* (Morphotype 1b), a not-elongated vertebra of a large 'dorudontid' basilosaurid (Morphotype 2) and 'shortened' vertebrae of a new, unnamed taxon (Morphotype 3). This article deals with a still undescribed, smaller vertebra, NMR-16642, from this site. Our first aim was to date it by dinoflagellate cysts in adhering sediments. Yielding an age of about 38 Ma, it is one of the very few remains of basilosaurids from Europe, of which the age could be assessed with reasonable certainty. The vertebra, Morphotype 1a, is assigned to a small species of *Pachycetus*. High-quality CT scans are used to differentiate between NMR-16642, Morphotype 1a, and the large species of *Pachycetus,* Morphotype 1b. Another aim of this paper is to investigate the inner structure and vascularity of the study vertebra and that of the other morphotypes (1b, 2, 3) from this area by using high-quality CT scans. Notwithstanding differences in size, shape and compactness, the vertebral inner structure with a multi-layered cortex of periosteal bone, surrounding two cones of endosteal bone appears to be basically similar in all morphotypes. Apparently, this inner structure reflects the ontogenetic vertebral growth. An attempt to reconstruct the vascularity of the vertebrae reveals a remarkable pattern of interconnected vascular systems. From the dorsal and, if present, ventral foramina, vascular canals are running to a central vascular node. From this node a system of vascular canals goes to the epiphyseal ends, giving rise to separate systems for cortex and cones. It is the first time that the vascularity of vertebrae of archaeocetes is investigated.

## INTRODUCTION

Brandt added to his review of the fossil cetaceans of Europe (1873) a description by Paulson regarding several large archaeocete vertebrae from Ukraine (called *Zeuglodon rossicus* by Paulson) and changed this name to *Zeuglodon paulsonii Brandt, 1873*. Torso vertebrae of this taxon are quite characteristic, in being (1) large (with a lumbar vertebral length of up to more than 280 mm), (2) anteroposteriorly elongated, and in having (3) a thick, multi-layered cortex, (4) a pock-marked surface, (5) anteroposteriorly elongated transverse processes, nearly as long as the centrum, (6) anteroposteriorly elongated pedicles of the neural arch and (7) pachyostotic pedicles of the neural arch (*Brandt, 1873*; *Gol'din & Zvonok, 2013*). Nowadays, the genus is named *Pachycetus* (*Van Vliet et al., 2020*), belonging to the subfamily Pachycetinae (*Gingerich, Amane & Zouhri, 2022*). Apart from the large *Pachycetus* remains from Europe, two small pachycetine species were described during the last decades: *Pachycetus wardii Uhen, 1999* from Bartonian strata of North Carolina (*Uhen, 1999*; *Uhen, 2001*) and Virginia (*Weems et al., 2011*), and *Antaecetus aithai Gingerich & Zouhri, 2015* from Bartonian sediments in Morocco and Egypt (*Gingerich & Zouhri, 2015*; *Gingerich, Amane & Zouhri, 2022*). Also from Spain a vertebra of a small species of *Pachycetus* has been described (*Van Vliet et al., 2023*).

The vertebra NMR-16642 in this study is part of a collection of fourteen vertebrae and one neural arch, which have been dredged between 1996 and 2017 from tidal channels Wielingen and Het Scheur in the Western Scheldt Estuary along the Belgian-Dutch border. They have basilosaurid characteristics and represent large taxa, obviously differing from each other and belonging to three morphotypes. All vertebrae but one were already described (*Post, 2007*; *Schouten, 2011*; *Post & Reumer, 2016*; *Post, Hoekman & Wilde, 2017*; *Van Vliet et al., 2022*). However, NMR-16642 remained undescribed. It is assigned to a small species of *Pachycetus*, Morphotype 1a in this article. It is not-abraded and was still firmly embedded in clayish sediments, which enabled dating by age-diagnostic dinoflagellate cysts. Being the first fossil of *Pachycetus* in Europe, which age could be precisely assessed, it is described separately in this study. NMR-16642 is significantly smaller and seems to be less osteosclerotic than most other vertebrae of *Pachycetus* from this site; therefore the inner structure of NMR-16642 is investigated with CT scans. In addition, the inner structure of three different types of large basilosaurid vertebrae from this site is investigated with CT scans and a systematic and detailed comparison is made, which had not been conducted before.

### Geological setting

Marine Palaeogene strata are covered by relatively thin beds of Neogene sediments in the area of the tidal channels Wielingen and Het Scheur in the Western Scheldt Estuary near the Belgian-Dutch border (Fig. 1). At Het Scheur, the Quaternary strata are less than 2.5 m thick (*Du Four et al., 2006*).

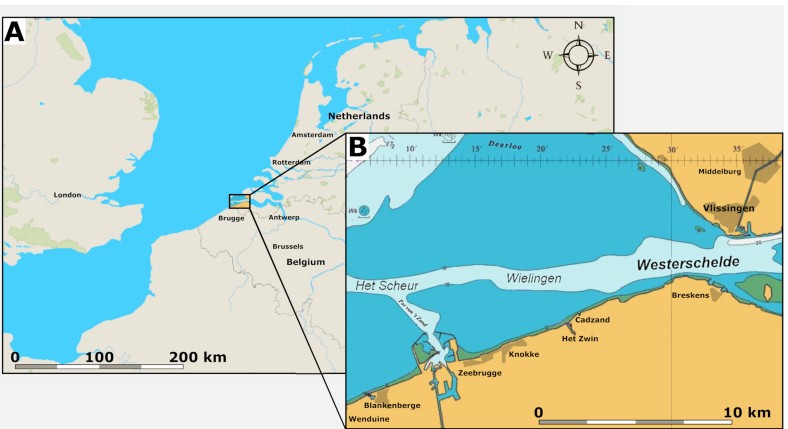

**Figure 1  Location of the fossil site.** (A) Map of Belgium and the Netherlands, modified after *Van Vliet et al.* (*2022*: fig. 1). (B) The location of the tidal channels Wielingen and Het Scheur at the Belgian-Dutch border.

These channels have been artificially deepened for navigation since the sixties of the last century and Palaeogene strata crop out at the seafloor (*Van Vliet et al., 2022*). At Het Scheur, the Middle Eocene Maldegem Formation (Lutetian to Bartonian in age), but at Wielingen, located more to the east, the Priabonian Zelzate Formation is encountered, because the Palaeogene strata are tilted and dip to the northeast (*Du Four et al., 2006*; *Post & Reumer, 2016*). The Maldegem Formation consists of grey and blue-grey, fine silts and clay, reaching here a thickness of about 45 to 60 m (*Du Four et al., 2006*). Seven members are distinguished in the Maldegem Formation (*Le Bot et al., 2003*) of which the Onderdijke Member, consisting of blue-grey clays and the Buisputten Member, consisting of sands are relevant for this study. Palaeogene sediments in Belgium show many hiatuses, indicative of transgressive and regressive phases (*Vandenberghe et al., 2004*) (Fig. 2).

# MATERIAL AND METHODS

The vertebral centrum NMR-16642 was dredged by commercial fishing vessel OD 31 in 2017. The vertebra is housed in the collections of Het Natuurhistorisch (the Natural History Museum Rotterdam, NMR) in the Netherlands.

*Aims of the study*—The analysis regarding the gross anatomy of vertebral centrum NMR-16642 by use of CT scans and the comparison with CT scans of the other three morphotypes of basilosaurid vertebral centra from this site. This concerns the bone structure and architecture, as well as the vertebral vascularity.

*Palynological analysis*—The dating is based on dinoflagellate cysts analysis of the sediments, adhering the vertebra. The accompanying sediment deposition is considered to be contemporaneous with the vertebra itself. Standard palynological techniques, including HCL and HF digestion, no oxidation and 15 μm sieving, were applied. The slides were mounted in glycerine jelly. Dinocyst taxonomy is according to that cited in the Lentin and Williams Index 2019 (*Fensome, Williams & MacRae, 2019*). One microscope slide

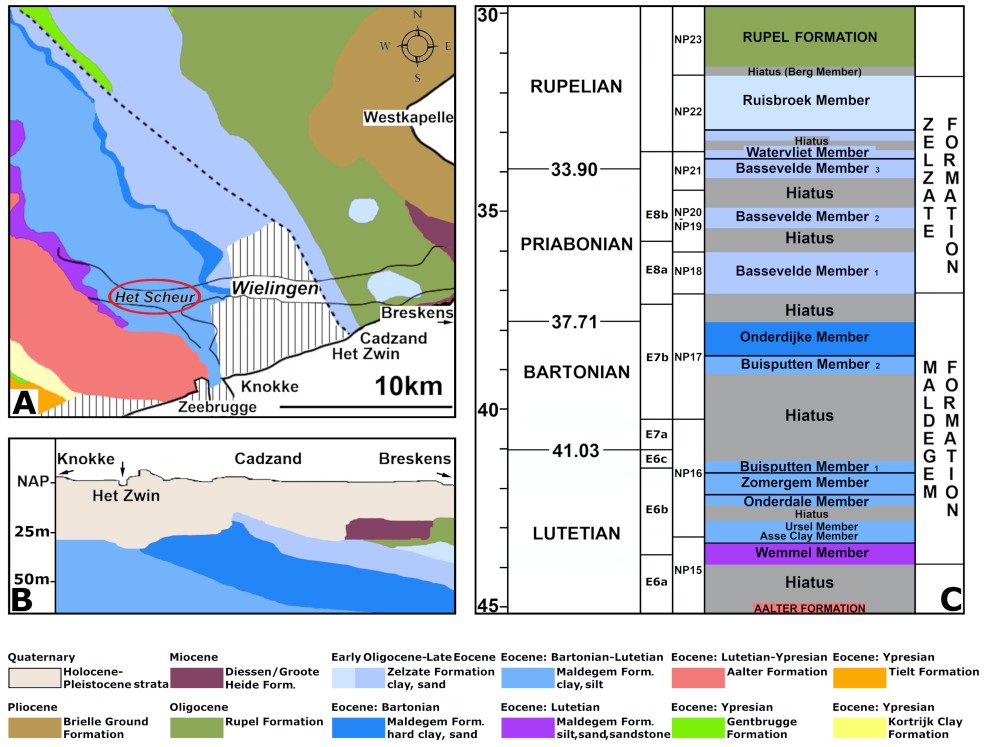

**Figure 2** **Section of the Eocene strata at the fossil site.** Stratigraphy of the Paleogene and Neogene sediments in the region of Wielingen/Het Scheur at the Belgian-Dutch border: (A) Paleogene and Neogene strata directly beneath the Pleistocene and Holocene sediments at the seafloor, modified after *Du Four et al. (2006*: fig. 3). The place where vertebra NMR-16642 has been found, is indicated with a red ellipsis. (B) The tilting of the Paleogene strata toward the north-northeast in the study area modified after *Du Four et al. (2006*: fig. 4). (C) Section of the Zelzate and Maldegem Formation, modified after *De Smet, Martens & De Breuck, (1997*: tabel 2.1); *Vandenberghe et al. (2004*: fig. 6); *Steurbaut et al. (2015*: figs. 3-4). Corrections for the ages after *Bujak & Mudge (1994)*, *Eldrett et al. (2004)* and *Gradstein et al. (2020*: Tables 28.1–28.2).

per sample was counted until a minimum of 200 palynomorphs (spores, pollen and dinoflagellate cysts) had been identified. The remainder of the slides were scanned for rare taxa. Miscellaneous fossils (like *e.g.*, *Pediastrum*, *Botryococcus*) were also counted, but kept outside the total sum of 200 specimens. The Eocene dinoflagellate cyst (dinocyst) zonation is based on *Bujak & Mudge (1994)*. This zonation is based on consistent dinocyst events (on last and first occurrence datums: LOD and FOD) from available peer-reviewed palynological contributions in NW Europe. The bio- and chronostratigraphy is adjusted after *Eldrett et al. (2004)* and *Gradstein et al. (2020)*.

*Measurements*—The relative length of a vertebral centrum is calculated by its dorsal anteroposterior length to anterior width ratio: relative length = Ld/Wa. Because the epiphyses are lacking in nearly all vertebral centra, the given relative length will be an underestimation. The relative width of a vertebral centrum is calculated by its anterior width to anterior height ratio: relative width = Wa/Ha.

The bone compactness of NMR-16642 is determined with the program 'Boneprofiler', which calculates the amount of bone in a given surface, which is the part occupied by bone

in this surface (*Girondot & Laurin, 2003*). To calculate the mean compactness of the several sections in a certain area, the sum of the surfaces occupied by bone is divided by the sum of the total investigated surface. As the images of the CT scan were slightly manipulated with Photoshop to sharpen the contrast between bone and intertrabecular space, the obtained values are an approximation of the real compactness.

*Vertebrae used for comparison*—The inner architecture of the vertebral centrum NMR-16442 is compared with the thirteen, earlier described vertebral centra from Het Scheur/Wielingen at the Belgian-Dutch border, all housed in the collections of Het Natuurhistorisch (the Natural History Museum Rotterdam, the Netherlands). Three different basilosaurid morphotypes are recognised (*Van Vliet et al., 2022*).

Morphotype 1a, anteroposteriorly elongated (torso) vertebrae (in which the length exceeds the width, having a relative length of 1.05 or more, and being significantly smaller than vertebrae of Morphotype 1b), represented besides NMR-16642 by a posterior thoracic or lumbar vertebral centrum NMR-150839 and by a caudal vertebra MSGB No. 25.191 from Spain, all ascribed to a small species of *Pachycetus.*

Morphotype 1b, elongated (torso) vertebrae (in which the length exceeds the width and having a relative length of 1.05 or more), represented by a central-posterior thoracic vertebral centra NMR-12331 and NMR-12332, as well as a lumbar vertebral centrum NMR-3404, all ascribed to a large species of *Pachycetus.*

Morphotype 2, not-elongated (torso) vertebra (in which the length equals the width and having a relative length of 0.75 to 1.05), represented by a large posterior thoracic or lumbar vertebral centrum, NMR-10284, an indeterminable basilosaurid;

Morphotype 3, 'shortened' (torso) vertebra (in which the length is smaller than the width and having a relative length of less than 0.75), represented by a caudal vertebral centrum NMR-10283, probably a new taxon, not comparable to any of the known basilosaurid genera (Fig. S1; Table S1).

Other vertebrae used for comparison:

MSGB No. 25.191, caudal vertebra of a small species of *Pachycetus,* from Taradell, Spain, housed in the collections of the Museo Geológico del Seminario de Barcelona in Spain (*Van Vliet et al., 2023*). This vertebra belongs to Morphotype 1a;

NsT90, posterior thoracic vertebra, *Pachycetus robustus Van Beneden, 1883* (holotype), housed in Museum für Mineralogie und Geologie, Senckenberg Naturhistorischen Sammlungen, Dresden (*Van Vliet et al., 2020*);

SMNS 10934b, large species of *Pachycetus* sp. from Gebel Mokattam, Cairo of Egypt, housed in the Staatliches Museum für Naturkunde Stuttgart, Germany. Both NsT90 and SMNS 10934b belong to Morphotype 1b.

USNM 510831, *Basilosaurus cetoides* (from a figure of *Houssaye et al., 2015*: fig. 14, without other information), housed in the National Museum of Natural History, Washington. This vertebra, being extremely anteroposteriorly elongated and less osteosclerotic than those of Morphotype 1b, could be called here Morphotype 1c.

*CT scan*—Scans were performed at the Department of Radiology and Nuclear Medicine, Erasmus Medical Center, Rotterdam, the Netherlands on a first generation photon-counting detector CT scanner (NAEOTOM Alpha, Siemens Healthineers, Erlangen,

Germany). Photon-counting CT typically can generate CT images at higher spatial resolution compared to conventional energy integrating detector CT, while being able to scan larger objects than micro-CT. For this study 0.2 mm overlapping cross-sectional slices were acquired with the ultra-high resolution scan mode allowing a resolution up to 0.11 mm (in-plane). From the primary scan data axial, coronal and sagittal 2D CT images were generated. 3D images were reconstructed on a post processing server (Syngo VIA VB60, Siemens Healtineers, Erlangen, Germany) with a "cinematic rendering technique". Cinematic rendering is a relative new technique for CT data which allows 3D images while maintaining sharpness and resolution, where the more commonly used "volume rendering technique" requires to smooth the 2D dataset with lower resolution reconstruction algorithms prior to 3D reformatting.

The reconstructions of the dorsoventral vascular system in vertebral centra of *Pachycetus* spp. have been made by a compilation of slices from these CT scans. These were 16 slices of NMR-16642, each 1.9 mm thick; 37 slices of NMR-12331, each 2.6 mm thick; 26 slices of NMR-12332, each 3.8 mm thick; 23 slices of NMR-3404, each 4.1 mm thick.

*Terminology*—We follow *De Buffrénil et al. (1990)*, *Houssaye et al. (2015)* and *Martínez-Cáceres, Lambert & De Muizon (2017)* for anatomical and osteological terms.

## Systematic paleontology

Order CETACEA *Brisson, 1762*
Clade PELAGICETI *Uhen, 2008*
Family BASILOSAURIDAE *Cope, 1868*
Subfamily PACHYCETINAE *Gingerich, Amane & Zouhri, 2022*
Genus PACHYCETUS *Van Beneden, 1883*

*Material*—One central-posterior thoracic vertebral centrum (NMR-16642) Figure 3.
*Description*—Vertebral centrum with the transverse processes and the pedicles of the neural arch preserved. The epiphyses are lacking, indicating that the specimen involved was not yet fully-grown. The anterior epiphyseal side is more or less oval-shaped; the posterior epiphyseal side is trapezoid-shaped. The multi-layered cortex is thick. The dorsal surface has a prominent medial ridge, running over the entire length of the vertebra. Several small foramina appear to be present on the dorsal surface. There are no ventral foramina. The pedicles of the neural arch are elongated, pachyostotic and massive as are the transverse processes. The transverse processes are directed more or less horizontally and are located at about the midpart of the centrum. They are stocky and the projection is short. Laterally, they bear a deep, oval fovea for the tuberculum of the rib which is confluent with a shallow fovea for the capitulum of the rib, located somewhat more anteriorly on the transverse process. Like in *Antaecetus aithai*, the deep foveae have a pitted surface, suggestive of a cartilaginous or ligamentous articulation with the rib (*Gingerich, Amane & Zouhri, 2022*) (Figs. 3E and 3G). The ventral side of the centrum is broad, wide and more or less flat. It has three antero-posteriorly directed ridges: a medial one and two lateral ridges. The

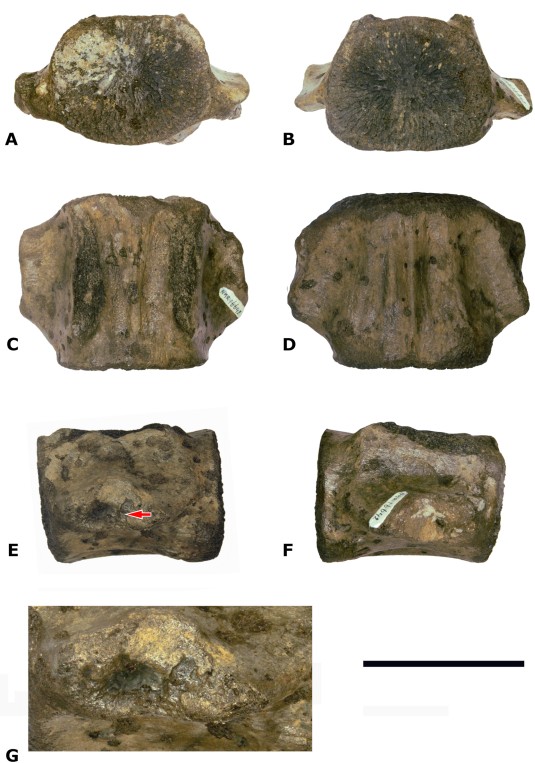

**Figure 3** **Vertebral centrum NMR-16442.** Morphotype 1a, central-posterior thoracic centrum, NMR-16442, small species of *Pachycetus* from Het Scheur at the Belgian-Dutch border, in anterior (A), posterior (B), dorsal (C), ventral (D), left lateral (E) and right lateral (F) view. (G) magnified part of the transverse process with the fovea for the rib. Arrow in E indicates the fovea for the rib. Scale bar in A–F is 100 mm; scale bar in G is 50 mm. See also Table S1.

lateral ridges start anteriorly near the medial ridge and widen sidewise to the posterolateral corners of the vertebral centrum. Here, they have flat protuberances, resembling hemal processes in caudal vertebrae. An elongate, but shallow and small fossa is located at both sides of the medial ridge. NMR-16642 has a dorsal length of 111 mm, an anterior width of 101 mm, and an anterior height of 80 mm. It is moderately elongated, with a length to width ratio of 1.10. The width exceeds the height, with a width to height ratio of 1.26. The maximum width including the transverse processes is 156 mm. The centrum is not or only slightly permineralised, and it is not abraded. It is grey-brownish in colour. There are multiple drilling holes, made by marine organisms.

*Assignment*—The pachyostotic and elongated pedicles of the neural arch, and the antero-posterior elongation of the transverse processes of NMR-16642 are considered diagnostic for Pachycetinae (*Kellogg, 1936*; *Gol'din & Zvonok, 2013*; *Gingerich & Zouhri, 2015*; *Van Vliet et al., 2020*). Because of the lateral foveae and the articulation pits for the ribs, NMR-16642 is considered a thoracic vertebra. The moderate vertebral elongation and the place of the transverse processes at the midpart of the lateral sides point to a central-posterior position (tentatively assigned to Th8-11). More anteriorly positioned thoracic vertebrae are less elongated, the transverse processes are placed more dorsally

(*Gol'din & Zvonok, 2013*: figs. 5–6) and have a smooth ventral side (*Van Vliet et al., 2020*). In posteriormost thoracic vertebrae the transverse processes are placed ventrally on the lateral sides (*Uhen, 2001*: fig.4).

# RESULTS

*Palynological analysis*—The palynological assemblage is relatively rich in marine dinoflagellate cysts. The variety of cysts is high (Fig. S2). Tasmanaceae (phrasinophyte algae associated with nutrient-rich, shallow marine conditions and stagnated ventilation) are also well represented. In Eocene strata Tasmanaceae are only commonly recorded in the Asse Clay Member (Dutch lithostratigraphical equivalent of the Maldegem Formation) from Zeeuws-Vlaanderen (*e.g.*, *Munsterman, 2003*; *Munsterman, 2004*). Reworking from a slightly older mid-Eocene, Lutetian stage is also recorded (*e.g.*, by the presence of the dinocysts *Areosphaeridium ebdonii, Diphyes pseudoficusoides, and Rhombodinium rhomboideum).* A late Middle Eocene, Bartonian age is inferred by the FOD of *Rhombodinium draco* and the LOD's of *Rottnestia borussica* and *Areosphaeridium fenestratum*, dinocyst zone E7b, last part of NP16 to mid NP17 (*Bujak & Mudge, 1994*; *Powell, 1992*; *Gradstein et al., 2020*). An origin from older strata, which have been eroded later, can be excluded, as the vertebra does not show any signs of abrasion by sea currents or transport. The current dating is confirmed by the presence of *Cordosphaeridium funiculatum* (FOD in the Bartonian) (Fig. 4). Large hiatuses are present in the local section (*Vandenberghe et al., 2004*). Based on the results of the dinoflagellate cysts interpretation, the vertebra could originate from two members of the Maldegem Formation, the Onderdijke Member or the Buisputten Member 2. The latter consists of sand (see 'Geological setting'), the former of stiff clay with some silts and sands (*Le Bot et al., 2003*). The vertebra was embedded in clay and, being not-abraded, indicating an origin from the late Bartonian Onderdijke Member as considered most probable. Hence the age is approximately about 38 Ma (37.7–38.6 Ma) (Fig. 4).

## Inner structure of NMR-16642

*Cortex*—The cortex of NMR-16642 consists of a thin, compact outer part and a thick, spongious-like, multi-layered inner part (Fig. 5; Figs. S3A and S4A). At the axial midpart of NMR-16642, the thin outer part is about 0.75 to 2.5 mm thick. The multi-layered inner part is maximum 17 to 20 mm thick in this section ('1' in Fig. 5B). Beneath the multi-layered cortex, spongious-like bone with a chaotic architecture is present. It is most prominent at the vertebra's midpart and within the transverse processes ('2' in Figs. 5B, 5D and 5F; Fig. S3A).

*Cones*—Two cones constitute a large part of the inner structure of the vertebral centrum. The top of the two cones meet each other at the vertebral midpart, but they are not visible in the axial section of the vertebral midpart (Fig. 5A). The cones consist of a seemingly amorphous spongious bone ('3' in Figs. 5D and 5F; Fig. S6A–S6B).

*Compactness*—The compactness is measured in axial sections through the midpart of the vertebra and through the anterior cone. In both sections, the compactness of the cortex (outer part and looser multi-layered part underneath) is higher than that of the central

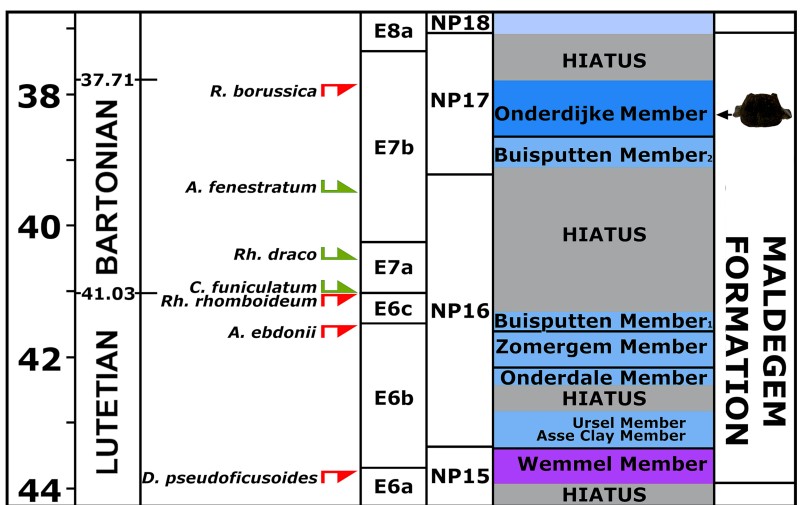

**Figure 4 Section correlated with encountered dinoflagellates.** Part of the section with age-diagnostic dinoflagellates in the sediments originally adhering to vertebra NMR-16642. For not-abbreviated names of dinoflagellates, see text.

parts. The highest cortical compactness is found at the lateral side of the vertebral centrum, near the outer part of the transverse process, and at the ventral side (about 0.84). Here, the vertebra can be called osteosclerotic (*De Buffrénil et al., 2010*). Ventrally and dorsally, the compactness of the cortex is somewhat lower. The compactness of the cone is higher, than that of the vertebral midpart. In both sections, the central part of the vertebra has the lowest compactness with values as low as 0.4 (Tables 1–2; Fig. S7).

*Vascularity*—Four, maybe five main vascular systems are discerned in NMR-16642: (1) a system of midvertebral dorsal vascular canals (the midvertebral VC), (2) a system of vascular canals surrounding the two cones, here called epiconal vascular canals (epiconal VC), (3) a system of vascular canals within the two cones, called endoconal vascular canals (endoconal VC), (4) a system of tiny vascular canals within the layers of the cortex, called endocortical vascular canals (endocortical VC) and (5) the apparently randomly scattered tiny vascular canals, called accessory vascular canals (accessory VC). The accessory VC are however hardly or not discernible in NMR-16642, probably due to their small size and the spongious-like structure of the cortex. The midvertebral VC and the epiconal VC are directly connected in a central point at the vertebra's midpart, here called the central vascular node (Fig. S5A). The vascularity of the epiphyseal discs has not been studied as the vertebra is lacking the epiphyses.

(1) *The midvertebral VC*—On the dorsal surface of vertebral centrum NMR-16642, six foramina are discerned, orifices of vascular canals. One central vascular canal is running from the left side of the dorsal surface in a straight line to the central vascular node. It bends dorsally slightly posteriorly. Its width varies between 8.0 mm at the dorsal surface to 4.5 mm at the central vascular node. The right central canal is partly obliterated which is probably an anomaly. Instead, three additional canals are seen, bending posteriorly to the dorsal surface, without a left counterpart. Additionally, anterior to the open central canal,

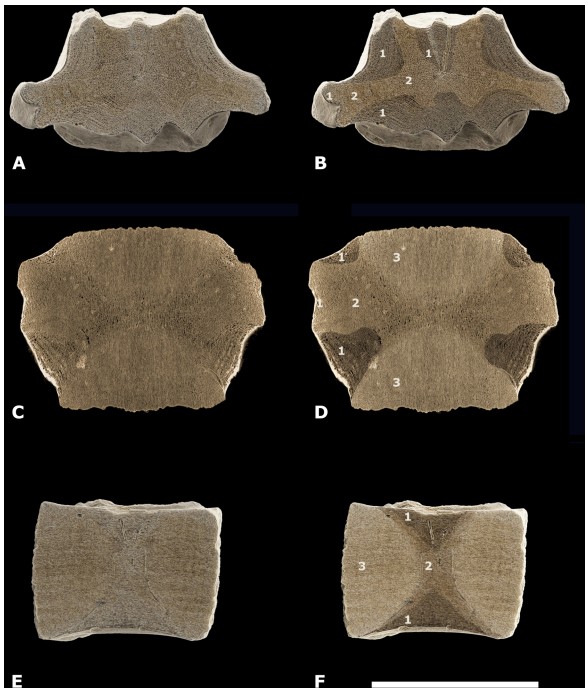

**Figure 5  Different types of bone in NMR-16642.** (A1-C2) CT scan images of the posterior thoracic vertebral centrum, *Pachycetus* sp., NMR-16442 from Het Scheur, at the Belgian-Dutch border, in axial (A–B), coronal - upper part of the centrum (C–D), and sagittal (E–F) sections. (B), (D), (F) have been artificially embrightened in colour, to indicate different parts of the spongious bone. Dark-brown parts show the multi-layered bone beneath the cortex (indicated by '1'), medium-beige parts show the more or less disorganised spongious bone underneath the multi-layered bone (indicated by '2'); light-yellowish parts show the cones (indicated by '3'). See also Table S1. Scale bar is 100 mm.

two canals arise from the left and right of the median ridge and fuse on their way to the central vascular node (Fig. 6A). NMR-16642 lacks ventral vascular canals or foramina.

(2) *The epiconal VC*—Along the surface of the cones in NMR-16642, a peculiar pattern of about 20-30 vascular canals is present, starting from the vascular node and running at first radially, then oblique and after that running longitudinally to the epiphyseal ends. Here they pierce the multi-layered cortex crossing the vascularised inner layers of the cortex. They are here called epiconal canals, surrounding the cones and resemble the ribs of an umbrella. They are schematically indicated in Fig. 6A. Some of these canals are rather wide, being more than 2.3 mm in diameter (Figs. S8A, S9A and S10A).

(3) *The endoconal VC*—The cones contain multiple longitudinal vascular canals going in a straight way from the epiconal VC toward the epiphyseal sides (Figs. S6A, S8A and S11A).

(4) *The endocortical VC*—Another vascular system is constituted by multiple tiny longitudinal blood vessels, present in all layers of the cortex (see 'Discussion'). They are running ventrally and laterally in a anteroposterior direction (Fig. S12A), but at the transverse processes they bend toward the tip of the transverse process (Fig. S13A). About

**Table 1  Compactness of vertebra NMR-16642 from het Scheur at the Belgian-Dutch border: vertebral midpart.**

| Region axial midpart | Section | Compactness (Co) | Surface (Su) (in cm) | Sum value (Co.Su) | Mean value |
|---|---|---|---|---|---|
| **Cortex lat** | | | | | |
| Dorsal left | **1** | 0.668 | 15 × 12 | 1.2024 | |
| Dorsal right | **2** | 0.704 | 12 × 15 | 1.2672 | |
| **1-2** | | | **3.6** | **2.4696** | **0.686** |
| **Cortex** | | | | | |
| Pr tr right | **3** | 0.841 | 12 × 15 | 1.5138 | |
| **3** | | | **1.8** | **1.5138** | **0.841** |
| **Cortex ventral** | | | | | |
| Left | 4 | 0.669 | 15 × 12 | 1.2042 | |
| Right | 5 | 0.68 | 19 × 16 | 2.0672 | |
| **4-5** | | | **4.84** | **3.2714** | **0.676** |
| **Central part** | | | | | |
| Dorsal left | **6** | 0.486 | 15 × 12 | 0.8748 | |
| Midpart right | 7 | 0.4 | 15 × 12 | 0.72 | |
| Midpart left | **8** | 0.455 | 15 × 12 | 0.819 | |
| Midline left | **9** | 0.578 | 15 × 12 | 1.0404 | |
| central midline right | **10** | 0.582 | 15 × 12 | 1.0476 | |
| **6-10** | | | **9** | **4.5018** | **0.5** |
| **Total compactness 1–10** | | | **19.24** | **11.7566** | **0.611** |

**Notes.**
First column: see Fig. S7. X-Y = sum of the values × to Y (total surface; total amount of bone in the total surface; mean value).
Second column: numbers correspond to the sections given in Fig. S7.
Third column gives the surfaces used for measurements, with (in bold) the total surface.
Fourth column: the sum value (Co.Su) gives the total amount of bone in the total surface used for measurements.
Fifth columns: the mean value is the total amount of bone divided by the total surface = the mean amount of bone.

145 canals in axial sections are present in one $cm^2$ of the ventral cortex. The canals are maximum about 1.5 mm, but mostly less than one mm in diameter at the vertebral midpart.

(5) *The accessory VC*—Canals of this system are hardly or not discernible in NMR-16642, probably due to their small size and the spongious-like structure of the cortex.

## COMPARISON

The inner architecture of the vertebral centrum NMR-16442, Morphotype 1a is compared with vertebral centra from Het Scheur and Wielingen at the Belgian-Dutch border, in which the following basilosaurid morphotypes are recognised:

Morphotype 1a, also represented by NMR-150839. Added to these MSGB No. 25.191 from Spain;

Morphotype 1b, represented by NMR-12331, NMR-12332 and NMR-3404. Added to these USNM 510831 from the USA; can be considered Morphotype 1c'.

Morphotype 2, represented by NMR-10284;

Morphotype 3, represented by NMR-10283. See also 'Materials and Methods', Fig. S1 and Table S1.

**Table 2  Compactness of vertebra NMR-16642 from het Scheur at the Belgian-Dutch border: anterior cone.**

| Region anterior cone | Section | Compactness (Co) | Surface (Su) (in cm) | Sum value (Co.Su) | Mean value |
|---|---|---|---|---|---|
| **Cortex & conus** | | | | | |
| Midpart lateral left | 1 | 0.897 | 1.5 × 1.2 | 1.6146 | |
| Ventral left | 2 | 0.828 | 1.5 × 1.2 | 1.4904 | |
| Midline ventral | 3 | 0.794 | 1.5 × 1.2 | 1.4292 | |
| **1-3** | | | **5.4** | **4.5342** | **0.84** |
| **Conus dorsal** | | | | | |
| Left | 4 | 0.774 | 1.5 × 1.2 | 1.3932 | |
| Midline | 5 | 0.656 | 1.5 × 1.2 | 1.1808 | |
| **4-5** | | | **3.6** | **2.574** | **0.72** |
| **Conus central** | | | | | |
| Midline left | 6 | 0.534 | 1.5 × 1.2 | 0.9612 | |
| Midline right | 7 | 0.625 | 1.5 × 1.2 | 1.125 | |
| Left | 8 | 0.715 | 1.5 × 1.2 | 1.287 | |
| **6-8** | | | **5.4** | **3.3732** | **0.625** |
| **Total compactness 1-8** | | | **14.4** | **10.4814** | **0.728** |

**Notes.**

First column: see figs. 1B and 2A. X-Y = sum of the values × to Y (total surface; total amount of bone in the total surface; mean value).

Second column: numbers correspond to the sections given in Fig. S7.

Third column gives the surfaces used for measurements, with (in bold) the total surface.

Fourth column: the sum value (Co.Su) gives the total amount of bone in the total surface used for measurements.

Fifth columns: the mean value is the total amount of bone divided by the total surface = the mean amount of bone.

*Dimensions of NMR-16642 and NMR-12332*—Vertebra NMR-12332, Morphotype 1b, large species of *Pachycetus*, is assigned a central-posterior thoracic position in the vertebral column (tentatively assigned to Th7-10), because of its relative length and place of the transverse processes. It is however much larger than NMR-16642 (tentatively assigned to Th8-11) and axial sections of NMR-16642 fall completely within the innermost cortical boundaries of NMR-12332 (Fig. 7). The smaller size of NMR-16642 cannot be attributed to a more anterior position than NMR-12332, because the transverse processes are placed more ventrally in NMR-16642 (Figs. 7C, 8B and 8E), indicating that it had a more posterior position in the vertebral column, than NMR-12332. Both NMR-16642 and NMR-12332 have a thick, multi-layered cortex. It is not conceivable that NMR-16642 represents an ontogenetic very young individual of the large *Pachycetus* species, which vertebrae would finally get dimensions as large as NMR-12332. Increase in size implies the deposition of ever more cortical bone layers and the internal morphological structure of NMR-16642 would become very different from that of NMR-12332, unless extensive remodelling of the innermost cortical layers would have taken place. No sign of this is noted in NMR-12332, in which the inner border of the multi-layered cortex is rather sharply delineated from the bone underneath. The small (NMR-16642) and the large (NMR-12332) vertebrae therefore must represent two different species of *Pachycetus*. The other representative of Morphotype 1A, the posterior thoracic or lumbar vertebra NMR-150839 from Het Scheur, is like NMR-16642 small in size, with a length of only 121 mm, an anterior (?) width of

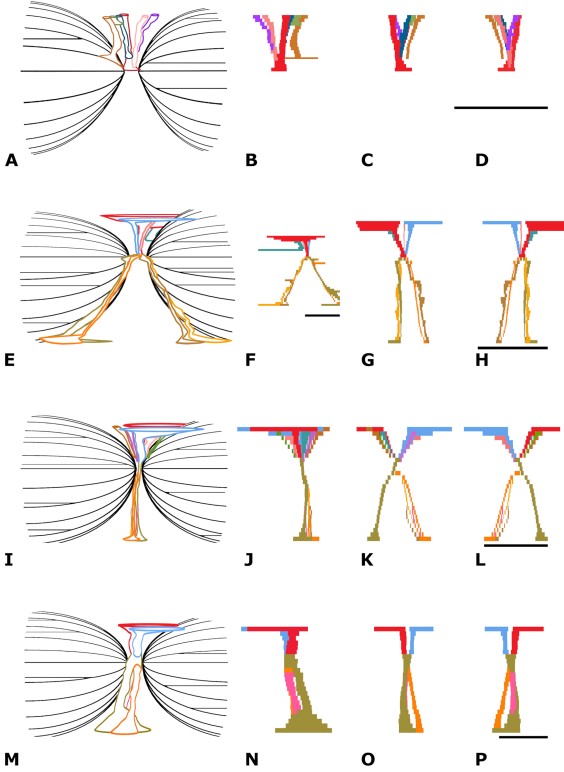

**Figure 6 Vertebral vascular canals (VC).** Comparison of the schematically figured midvertebral, epi-conal and endoconal VC within vertebral centra, Morphotype 1a, small species of *Pachycetus* and Morphotype 1b, large species of Pachycetus, in right lateral (A, E, I, M), left lateral (B, F, J, N), posterior (C, G, K, O) and anterior (D, H, L, P) view. (A–D) Morphotype 1a, central-posterior thoracic vertebral centrum, NMR-16642. (E–H) Morphotype 1b, central-posterior thoracic vertebral centrum, NMR-12331. (I–L) Morphotype 1b, central-posterior thoracic vertebral centrum, NMR-12332. (M–P) Morphotype 1b, lumbar vertebral centrum, NMR-3404. In NMR-16642, ventral midvertebral VC are lacking. Multiple, rather small dorsal and ventral midvertebral VC are seen in NMR-12331 and NMR-12332. Only a few, but large midvertebral VC, ending in large fossae are seen in the lumbar NMR-3404. Colours used have no other meaning than to differentiate between the several midvertebral VC. See also Table S1. Scale bar is 50 mm.

102, and an anterior (?) height of 89. Its relative length is 1.19 (without epiphyses). The vertebra lacks a thick compact cortex (Figs. 9C–9D).

A caudal vertebra, MSGB No. 25.191 from Taradell, Spain, with an estimated original length of 138 mm, was also assigned to an European small species of *Pachycetus* (*Van Vliet et al., 2023*) (Fig. 10B). The dimensions of NMR-16642 equal those of the posterior thoracic vertebrae of *Pachycetus wardii* (*Uhen, 1999*) (missing one or two epiphyses, see (*Uhen, 2001*)) from North Carolina, or *Antaecetus aithai* from Morocco (*Gingerich, Amane & Zouhri, 2022*). Grooves and ridges at the ventral side of NMR-16642 are similar to a presumably posterior thoracic vertebral centrum (NsT90) of *Pachycetus robustus* from Germany (*Van Vliet et al., 2020*). However, NMR-16642 is smaller in size than NsT90 (Fig. 8A–8B, resp. Fig. 8C–8D). NMR-16642 is also significantly smaller than the central-posterior thoracic vertebrae of the large species of *Pachycetus* known from Ukraine (*Kellogg, 1936*: tables 24 and 25), Germany (*Uhen & Berndt, 2008*; *Van Vliet et al., 2020*; *Gingerich,*

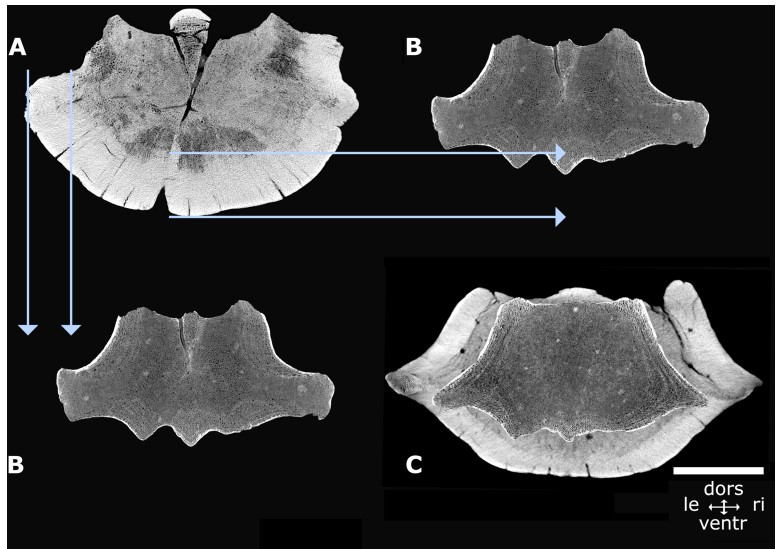

**Figure 7 Axial cross sections NMR-16642 and NMR-12332.** Axial cross sections of the vertebral centrum NMR-16642 compared with that of the vertebral centrum NMR-12332, showing that NMR-16642 completely falls within the inner boundaries of the multi-layered cortex of NMR-12332. This indicates that NMR-16642 is from a different, smaller species of *Pachycetus* than the large *Pachycetus* sp., NMR-12332. (A) Morphotype 1b, NMR-12332. (B) Morphotype 1a, NMR-16642 (A-B axial sections through the vertebra's midpart). (C) Projection of NMR-16642 upon NMR-12332 (axial section through the anterior cone). Blue arrows indicate the external and internal boundaries of the multi-layered cortex of NMR-12332. See also Table S1. Scale bar is 50 mm.

*Amane & Zouhri, 2022*) and from Belgium (*Van Vliet et al., 2022*). See Table S1 and Fig. S14 for a comparison of the dimensions of central to posterior thoracic vertebrae, assigned to small and large species of *Pachycetus*.

## Inner structure of the vertebrae used for comparison

*Cortex*—Morphotype 1a-b, elongated vertebrae. The multi-layered cortex in vertebrae of *Pachycetus* is ventrally much thicker than dorsally (Figs. S3A–S3C). The cortex in the small vertebra NMR-16642, Morphotype 1a, consisting of a rather loose spongious-like bone, clearly differs from the very compact cortex seen in vertebrae NMR-12331, NMR-12332 and NMR-3404 of a large species of *Pachycetus* Morphotype 1b, in which the layering in the very compact outer part of the cortex is hardly visible. However, in Morphotype 1b, the inner layers consist of spongious-like bone. Notwithstanding the greater cortical compactness, the axial section of the midpart of NMR-3404 (Fig. 3C) is more or less similar to that of NMR-16642 (Fig. 5B). The cortex in vertebrae of Morphotype 1b is divided into two parts: a very compact one (1a) and a less compact one (1b).The transition between the two parts is more or less fluent, whereas the transition to the amorphous inner part is rather sharply delineated. *Davydenko, Tretiakov & Gol'din* (*2023*: p. 5) discerned three parts in the cortex of vertebrae of *Basilotritus* (*Pachycetus*) sp.: a compact outer part, a transition zone and a less compact inner part.

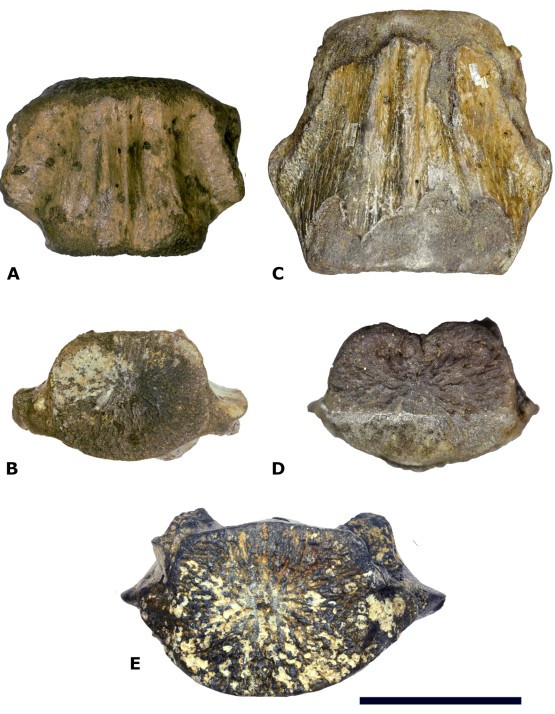

**Figure 8 Comparison of NMR-16442 with NsT90 and NMR-12332.** (A–B) Morphotype 1a, central-posterior thoracic vertebral centrum, NMR-16442, small species of *Pachycetus* from Het Scheur at the Belgian-Dutch border. (C–D) Morphotype 1b, posterior thoracic vertebral centrum, NST90, *Pachycetus robustus* (holotype), from the Helmstedt region, Germany in ventral (A, C) and anterior (B, D) view, originally described by *Van Beneden (1883)*, modified after *Van Vliet et al. (2020*: plate 2 C1, C4). (F) Morphotype 1b, central-posterior thoracic vertebral centrum, NMR-12332, *Pachycetus* sp. from Het Scheur, Belgian-Dutch border, in anterior view, modified after *Van Vliet et al. (2022*: fig. 11A). Both vertebrae NMR-16642 and NMR-12332 have pronounced ridges on the ventral side. The transverse processes in NMR-16642 are placed lower on the vertebra's lateral sides and protruding more laterally, than in NMR-12332, indicating a more posterior position in the vertebral column. See also Table S1. Scale bar is 100 mm.

Morphotype 2, not-elongated vertebrae. The large vertebral centrum NMR-10284 has a thick multi-layered cortex, which is dorsally much thicker, than the cortex of the elongated vertebral centra of Morphotype 1a-b. The cortex of NMR-10284 differs from that of the large *Pachycetus* in consisting of spongious-like, compact and less compact layers. At the left ventral side, the cortex is broken off through a spongious, more loose layer along a compact layer, a phenomenon in archaeocete vertebrae which was already observed by *Müller (1849)* (Fig. S3E).

Morphotype 3, 'shortened' vertebrae. Although the vertebra NMR-10639 is large, it consists entirely of spongious-like bone, except for a thin outer cortex. The layering of the thick, inner cortex is not clearly visible at most places, because of the loose structure. The cortex is dorsally only slightly thinner than ventrally (Fig. S3F).

*Cones*—All here described vertebral centra have two cones (Fig. S5).

Morphotype 1. Whereas the cones of NMR-16642, Morphotype 1a, consist of loose spongious bone (Figs. S5A and S6A–S6B), the cones in vertebrae of the large *Pachycetus*

Peer

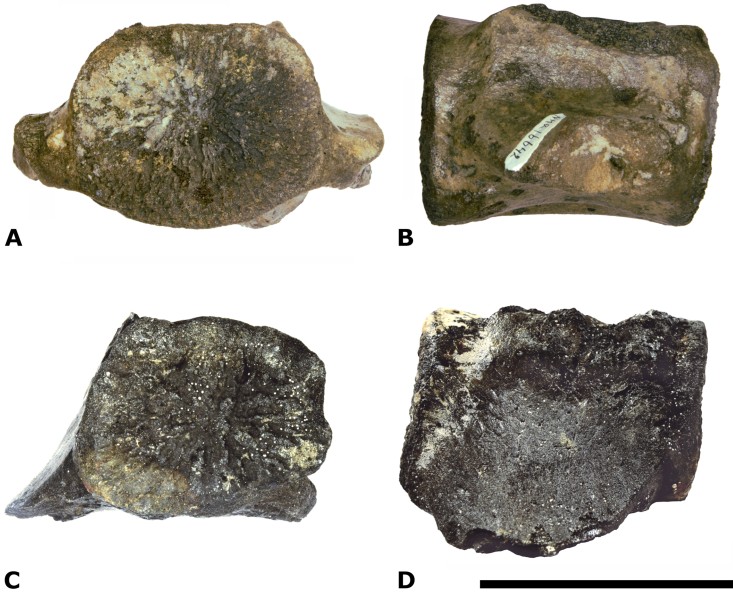

**Figure 9** **Comparison of NMR-16442 with NMR-150839.** Morphotype 1a, central-posterior thoracic vertebral centrum, NMR-16442, compared with posterior thoracic or lumbar vertebral centrum NMR-150839, both assigned to a small species of *Pachycetus* from Het Scheur at the Belgian-Dutch border. (A-B) NMR-16642 small species of *Pachycetus* from Het Scheur, Belgium, in anterior (A) and right lateral (B) view. (C-D) NMRT-150839 , in anterior (C) and right lateral (D) view. C, D Modified after *Van Vliet et al.* (*2022*: fig. 4A and 4F). See also Table S1. Scale bar is 100 mm.

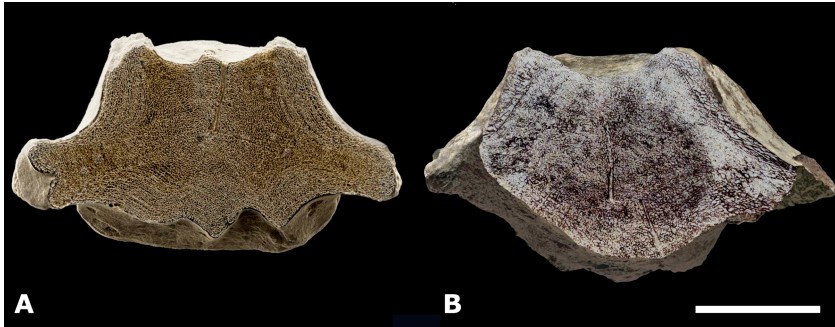

**Figure 10** **Comparison of the axial cross sections of NMR-16642 with MGSB No. 25.191.** (A) Morphotype 1a, central-posterior thoracic vertebral centrum NMR-16642, small species of *Pachycetus* from Het Scheur at the Belgian-Dutch border. (B) Morphotype 1a, caudal vertebral centrum, MGSB No. 25.191, small species of *Pachycetus* from Taradell, Catalonia, Spain, in axial cross-section with the section as figured by *Pilleri* (*1989*: plate II) superimposed; modified after *Van Vliet et al.* (*2023*: fig. 4C). Both NMR-16642 and MGSB No. 25.191 have a thick, multi-layered cortex surrounding not-layered bone; both vertebrae have a low compactness . Note that in (A) bone is white, cavities are black; in (B) bone is brown, cavities are white. See also Table S1. Scale bar is 100 mm.

sp., Morphotype 1b (NMR-12331, NMR-3404) are more compact, being less radiolucent than NMR-16642 in the CT scans (Figs. S5B–S5C and S6C–S6D). The spongious and very elongated cones of the large basilosaurine *Basilosaurus cetoides* clearly differ from the cones

of the investigated morphotypes in shape, being obtuse at the vertebral midpart (*Houssaye et al., 2015*; *Gingerich, Amane & Zouhri, 2022*).

Morphotype 2. The cones in the vertebral centrum NMR-10284 consist of a less compact spongious bone, than seen in vertebrae, Morphotype 1b of the large species of *Pachycetus* (Figs. S5E and S6E).

Morphotype 3. Notwithstanding the loose structure of the large NMR-10283, there are two cones, which are rather flat. (Figs. S5F and S6F).

Remarkably, the cones in all morphotypes are layered (Fig. S6). This indicates a not-continuous growth with dense and less dense bone depositions, reflecting differences in growth rate.

*Compactness*—Morphotype 1a. The compactness of vertebra NMR-150839 seems low, as it consists of spongious-like bone without a thick, compact cortex.

Morphotype 1b. The compactness of vertebrae of the large species of *Pachycetus* from Het Scheur at the Belgian-Dutch border, is apparently much higher than that of NMR-16642, Morphotype 1a, with a compact to very compact cortex, being white in the CT scans.

Morphotype 2. CT scans of NMR-10284 show a much higher overall compactness, than that of NMR-16642.

Morphotype 3. CT scans of NMR-10283 show a low compactness, resembling that of NMR-16642.

*Vascularity*—The vascular systems, described in NMR-16642, Morphotype 1a are discerned in all of the three other investigated morphotypes. The vascularity of the epiphyseal discs has not been studied as most vertebrae are lacking the epiphyses.

(1) *The midvertebral VC*—Morphotype 1b. This vascular system is investigated in detail in the central-posterior thoracic vertebral centra NMR-12331 and NMR-12332, and in the lumbar vertebral centrum NMR-3404 of the large species of *Pachycetus*. Contrary to the NMR-16642, ventral midvertebral vascular canals are present in vertebrae NMR-12331, NMR-12332, NMR-3404 (Figs. 6E–6P).

NMR-12331 has two main dorsal central canals, arising from separate fossae (left: 95 mm; right: 63 mm in length). Running to the central vascular node, they decrease in size (about 2 to 6.0 mm in width). They are accompanied by a tiny and a large canal. The large one does not go to the central vascular node, but to a epiconal vascular canal. Ventrally, four vascular canals, 1.5 to 9.0 mm in width, are present, a left–right anterior and a left–right posterior pair. Instead of going in a straight line to the ventral surface, they follow the course of epiconal VC, each ending in a separate narrow fossa of about 50 mm in length. They seem to be enlarged epiconal canals (Fig. 6E–6H).

NMR-12332 has two main dorsal vascular canals arising from separate fossae (left: 50 mm; right: 65 mm in length). Running to the central vascular node, they decrease in size (4.0 to 7.0 mm in width). The central canals are accompanied by several canals, bending posteriorly and anteriorly and all directed to the central vascular node. Ventrally, two vascular canals are present. The left one is about two mm in width, ending in a small ventral fossa, about 15 mm in length. The right one splits into four tiny canals, each about 1.0 to 1.5 mm in width, ending in in a ventral fossa about 13 mm in length (Fig. 6I–6L).

NMR-3404 has only two dorsal vascular canals arising from separate fossae (left: 61 mm; right: 56 mm in length). Running to the central vascular node, they decrease in size (each slightly more than 11 mm in width). There are no accompanying smaller canals. Ventrally, a large ventral vascular canal is positioned at the left side, that splits into a posterior (7.5 mm in width) and an anterior canal (nine mm in width). The anterior canal immediately splits into a left and a right canal (9.0 resp. 7.0 mm in width). The left canal fuses with the posterior canal ventrally. Near the ventral surface, the posterior canal and right anterior canal become larger, ending in two separate ventral fossa's (left: 58 mm; right: 28 mm in length) (Fig. 6M–6P).

Morphotype 2. NMR-10284 has four vascular canals, running from the dorsal surface to the central vascular node. The largest canal at the left side, is 6.5 mm in width; the largest one at the right side is 6.0 mm in width. The canals fuse near the central vascular node. Ventrally, three vascular canals are at the left side, the largest 4.2 mm in width; at the right side one large, 9.1 mm wide canal, is present. Near the ventral surface they are accompanied by several smaller canals.

Morphotype 3. NMR-10283 has very large midvertebral vascular canals (*Van Vliet et al., 2022*). Near the dorsal surface, two oval-shaped vascular canals are present; the left one and right one are maximum 16.2 mm, resp. 22.0 mm in width. They fuse toward the central vascular node. Near the ventral surface two oval-shaped vascular canals are present; the left one and the very large right one are maximum 9.0 mm reps. 31.0 mm in width.

(2) *The epiconal VC*—Morphotype 1b. The umbrella-shaped epiconal VC are figured in vertebra NMR-12331, the large species of *Pachycetus* (Figs. S8B, S9C and S10B). They are clearly noted on a partial cone, detached from a lumbar vertebra of *Pachycetus*, large sp., ID 20-4 from the Helmstedt region, Germany, as well as in a vertebral fragment, SMNS 10934b, of *Pachycetus* sp., large species, from the Bartonian of Egypt (Fig. S10C, resp. Fig. S10D). It was earlier described and figured by *Van Vliet et al.* (*2020*: p. 129 & plate 3 b1-3) and *Van Vliet et al.* (*2022*: p. 21 & fig. 14F) in large species of *Pachycetus*.

Morphotype 2, resp. 3. The epiconal VC are also present in vertebrae NMR-10284 and NMR-10283 (Fig. S10E, resp. Fig. S10F). The canals are about 0.5 to more than 1.5 mm in diameter.

(3) *The endoconal VC*—In the cones of all three morphotypes from Wielingen/Het Scheur, the longitudinal endoconal VC are running in a straight line toward the epiphyseal sides. Here they end between the ridges on the epiphyseal end of the vertebra. The diameter of the endoconal VC differs slightly in the three investigated morphotypes (Fig. S11).

(4) *The endocortical VC*—Morphotype 1b. In vertebra NMR-12331, the mostly longitudinal endocortical VC are only noted in the innermost layers of the cortex.

Morphotype 2. In vertebra NMR-10284 the endocortical VC are present in the innermost layers, as well as in several layers of the outer regions of the cortex (Fig. S12C–S12E).

Morphotype 3. In vertebra NMR-10283 the endocortical VC are present in all layers of the cortex, resembling NMR-16642 (Fig. S12F). The endocortical VC are similar in diameter in all three morphotypes and are running in the same directions (Figs. S12–S13). The innermost layers of vascular canals are sharply delineated from the amorphous inner part in especially the morphotypes 1a-b and 2 (Fig. S3).

(5) *The accessory VC*—These canals differ from the epiconal VC in running in a straight line and in not being curved around the cones. The accessory VC are generally smaller than the epiconal VC (less than 2.0 mm, resp. more than 2.3 mm in width), but are nevertheless sometimes difficult to tell them apart, especially when only a part of an accessory VC is visible in axial sections.

Morphotype 1b. The multiple, mostly tiny, radial accessory VC arise from the surface and are running toward the vascular node. Contrary to NMR-16642, Morphotype 1a, they are clearly seen at the ventral side of the compact cortex of NMR-12331.

Morphotype 2. In NMR-10284, the accessory VC appear to be abundant.

Morphotype 3. In NMR-10283 only some accessory VC are visible (Fig. S15).

## DISCUSSION

The architecture is basically similar in the here investigated basilosaurid vertebral centra, with two cones surrounded by a multi-layered cortex thickest at the midpart of the centrum. The presence of cones in vertebrae of *Pachycetus* and *Eocetus* reported by *Gingerich & Zouhri (2015)* as unusual, appears to be a common feature in basilosaurids. Also the presence of a compact, multi-layered cortex in vertebrae of *Basilosaurus cetoides,* reported by *Houssaye et al. (2015)* as unusual, is seen in vertebrae of both Morphotype 1b (large species of *Pachycetus*) and Morphotype 2 and cannot be considered uncommon in basilosaurid vertebrae.

In the three morphotypes, the architecture is a result of the dynamics of osteogenic processes. Like in all mammals, two types of bone are discerned: periosteal bone, deposited by the periosteum and giving rise to the cortex, and endochondral bone, deposited by the secondary growth centra at the epiphyseal discs and giving rise to the cones. Generally, both types of bone can be replaced and remodelled to spongious bone (*De Buffrénil et al., 1990*).

*Cortex*—Similar as in archaeocete ribs (*Gray et al., 2007*), in the here investigated basilosaurid vertebrae the multi-layered vertebral cortex consists of periosteal, cyclical bone depositions (lamellar-zonal bone *sensu De Buffrénil et al., 1990*). At the primary growth centrum in the midpart of the vertebral centra, the highest count of periosteal layers is deposited and the multi-layered cortex is thickest (Fig. S5). Toward the epiphyseal sides of the vertebrae, less and less periosteal bone layers are deposited, because here bone deposition occurs later in the individual's ontogenetic age. Remodelling and restructuring of bone with the appearance of secondary osteons, was seen in thin sections of the inner cortical layers of vertebra, NMNH-P Ngr-12, a large species of *Basilotritus* (*Pachycetus*) sp., from Ukraine (*Davydenko, Tretiakov & Gol'din, 2023*). However, in the here investigated vertebrae, the persisting layering of the inner parts of the cortex indicates that remodelling of periosteal bone was not complete, but reduced, similar to what is observed in ribs of *Basilosaurus cetoides* or *Zygorhiza kochii* (*De Buffrénil et al., 1990*). The spongious-like, not-layered bone, present beneath the multi-layered cortex differs from the cones and is therefore considered to be also periosteal bone. The absence of layering could indicate an extensive remodelling, but it is also possible that this noncyclical bone is deposited at

a very young ontogenetic age, when the animal is supplied with maternal nutrients. It is most prominent at the vertebra's midpart and within the transverse processes ('2' in Figs. 5B, 5D, 5F; Fig. S3C).

*Cones*—The cones obviously have been deposited by the secondary growth centra at the epiphyseal disc and consist of endochondral bone. In especially the anteroposteriorly elongated vertebrae of Morphotype 1a-b, the parabolic outlines of the cones reflect the at first rapid, but later on decelerating growth in diameter, as related to the growth in length of the vertebra. The layering of the spongious bone in the cones (Figs. S6A–S6D), as seen in particularly vertebrae of the small and large species of *Pachycetus*, suggests that the spongious bone of the cones underwent only moderate remodelling, contrary to bones in most mammals (*De Buffrénil et al., 1990*). In the morphotypes 2 and 3, only some lines are seen, remnants of layers of more compact spongious bone (Figs. S6E–S6F). This probably suggests that resorption and remodelling of bone here was more pronounced than in *Pachycetus* spp. (Morphotype 1a, b). Investigation of the microscopic bone structure will be necessary to confirm this hypothesis.

*Compactness*—With a significant lower compactness and a smaller size than torso vertebrae of large species of *Pachycetus*, NMR-16642 is assigned to a separate pachycetine taxon. NMR-150839 from the same site, has apparently a similar low compactness and most probably belongs to the same taxon. The caudal vertebra, MSGB No. 25.191 from Taradell, Spain, with the same characteristics, can possibly added to these. The compactness of the axial midpart of NMR-16642 is quite similar to that of the axial midpart of MSGB No. 25.191: the total compactness of the axial midpart of NMR-16642 is 0.61, that of the Taradell vertebra 0.56. The compactness of the multi-layered cortex of MSGB No. 25.191 is only slightly lower than that of NMR-16642 (0.53−0.58, resp. 0.68−0.69), whereas the compactness of the central part of MSGB No. 25.191 is slightly higher (0.55 resp. 0.50) (*Van Vliet et al., 2023*) (Fig. 10). A vertebra from Barton, housed in the NHM of London appears to be quite similar and will be described in a future study.

In vertebrae, Morphotype 1b, of the large species of *Pachycetus*, the cones are well delineated from the compact layers of the cortex and cones can easily become detached from the vertebral centrum (see *Van Vliet et al., 2020*). In vertebrae, Morphotype 1a, the compactness of the multi-layered cortex of NMR-16642 and MSGB No. 25.191 from Taradell, Spain (*Van Vliet et al., 2023*) is however not much higher than that of the cones and detachment of the cones will not easily happen. NMR-16642 and probably NMR-150839, as well as MSGB No. 25.191, seem to differ from vertebrae of the small pachycetine *Antaecetus aithai*, in which isolated cones were described by *Gingerich & Zouhri (2015)* and could represent a different taxon. Investigation of the compactness in vertebrae of *A. aithai* and *P. wardii* is necessary to confirm possible differences with the European small species of *Pachycetus*.

*Vascularity*—In thoracic vertebrae, Morphotype1a-b, the midvertebral VC are small and numerous. NMR-16642 lacks ventral vascular canals. In the lumbar vertebrae NMR-3404 (Morphotype 1b) and NMR-10284 (Morphotype 2), vascular canals are fewer in numbers, but larger. The caudal vertebra NMR-10293 (Morphotype 3) has only two dorsal and two ventral canals which are large to very large.

In vertebra NMR-16642, Morphotype 1a, small species of *Pachycetus*, the entire multi-layered cortex (whether originally compact or less compact) seems to have been replaced by spongious-like bone, but the original layering is still present. Small cavities are more or less regularly ordered according to these layers. They are nearly similar in size, contrary to the irregular cavities in trabecular bone. We interpret these cavities as vascular canals (endocortical VC), because complete remodelling and replacement of periosteal bone by trabecular bone cannot take place without destroying the layered architecture. Also in vertebrae of morphotypes 1b and 2, the cavities in alternating cortical layers of loose, spongious-like and compact bone, are interpreted as vascular canals (endocortical VC).

The endocortical VC were described and figured by *Davydenko, Tretiakov & Gol'din* (*2023*: fig. 6C) in microscopic sections of the cortex of lumbar vertebra NMNH-P Ngr-12, *Basilotritus* (*Pachycetus*) sp.(here: Morphotype 1b) as 'numerous longitudinal vascular canals distributed in circular rows' and 'a well-ordered structure of circular canals' in the deepest area of the cortex.

The accessory VC were described and figured by *Davydenko, Tretiakov & Gol'din* (*2023*: figs. 1B and 2A) in thoracic vertebra NMNH-P OF-2096, *Basilotritus* (*Pachycetus*) *uheni* and lumbar vertebra NMNH-P Ngr-12, *Basilotritus* (*Pachycetus*) sp.(both here Morphotype 1b) as 'radial vascular canals'.

It is interpreted that the blood supply to the central vascular node at the primary growth centrum is provided by the midvertebral VC and probably the accessory VC. This vascular node gives in its turn rise to the two umbrella-like epiconal VC. They surround the cones (Figs. S10C–S10D) and apparently grow in length together with the cones. The epiconal VC give rise to the endoconal VC and add to the blood supply of the epiphyseal ends from the inner side of the vertebral centrum (Fig. S11). By crossing the inner layers of the cortex (Figs. S10A–S10B, resp. Figs. S10E–S10F), the epiconal VC are possibly connected to the endocortical VC. It is presumably especially the orifices of the accessory VC, as well as those of the epiconal VC near the epiphyseal sides, that can be seen as pock marks on the vertebra's surface.

## CONCLUSIONS

The vertebral centrum, NMR-16642 dredged from the Western Scheldt Estuary at the Belgian-Dutch border in 2017, most probably originates from the Onderdijke Member of the Bartonian Maldegem Formation (Belgian Nomenclature). Analysis of the microfossil content of the adhering sediments reveals an age of ca. 38 Ma.

NMR-16642 is assigned to a small species of *Pachycetus* sp. It has a remarkable low compactness as compared to vertebrae of large species of *Pachycetus* from Europe. A posterior thoracic or lumbar vertebra, NMR-150839 from this site can be probably assigned to the same taxon, and maybe also a caudal vertebra from Taradell, Spain, equally small in size and with a similar low compactness.

Comparison of CT scan images of NMR-16442 with basilosaurid vertebrae of three morphotypes, all from Het Scheur and nearby Wielingen, reveals that their vertebral architecture is basically similar, consisting of a multi-layered cortex surrounding two

cones. Contrary to earlier observations, the presence of a thick, multi-layered cortex in the vertebral midpart of *Basilosaurus* is not an exception, but the common condition in the here investigated basilosaurid vertebrae. Moreover, the presence of cones is not restricted to the genera *Pachycetus* and *Eocetus*.

Apart from the not-investigated blood supply of the epiphyses, five vascular interconnecting systems are discerned within the basilosaurid vertebral centra, which for the first time are described in detail. Investigation of microscopic sections is necessary to validate the findings regarding bone structure and vascularity.

**Institutional abbreviations**

| | |
|---|---|
| **FSAC Bouj** | Faculté des Sciences Ain Chock, Boujdour collection, Université Hassan-II de Casablanca, Morocco |
| **KOM** | Kirovograd Oblast Museum, Ukraine |
| **MGSB** | Museo Geológico del Seminario de Barcelona, Spain |
| **NCSM** | North Carolina Museum of Natural Sciences, Raleigh, North Carolina, USA |
| **NMNH-P** | Paleontological Museum, National Natural History Museum of the Academy of Sciences of Ukraine, Kiev, Ukraine |
| **NHM** | Natural History Museum London, England |
| **NMR** | het Natuurhistorisch, Rotterdam (Museum of Natural History, Rotterdam), the Netherlands; |
| **NsT (=** | Museum für Mineralogie und Geologie: Niedersachsen Tertiär, Senckenberg |
| **MMG: NsT)** | Naturhistorischen Sammlungen, Dresden SMNS, Staatliches Museum für Naturkunde in Stuttgart (State Museum of Natural History in Stuttgart) |
| **USNM** | United States National Museum of Natural History, Washington DC, USA |

**Anatomical abbreviations**

| | |
|---|---|
| **ant** | anterior |
| **Co** | compactness |
| **post** | posterior |
| **Su** | surface |
| **VC** | vascular canals |

## ACKNOWLEDGEMENTS

We are grateful to Drs Bram Langeveld, curator of Het Natuurhistorisch (Natural History Museum, Rotterdam), the Netherlands, who welcomed us always in a very friendly way and gave us ample opportunity to study the vertebral centra from Wielingen/Het Scheur. Also we are grateful to Dr. Eli Amson who was very helpful during our visit to the Staatliches Museum für Naturkunde Stuttgart, Germany, to Dr. Mark Uhen who, very kindly, gave us a second opinion regarding vertebrae NMR-16642 and to Dr. Olivier Lambert who gave important observations on vertebra NMR-16642. We thank Dr. Davydenko and two anonymous reviewers for their ciritcal comments, which substantially improved the article.

### Funding

The authors received no funding for this work.

### Competing Interests

The authors declare there are no competing interests.

### Author Contributions

- Henk Jan van Vliet conceived and designed the experiments, analyzed the data, prepared figures and/or tables, authored or reviewed drafts of the article, and approved the final draft.
- Mark EJ Bosselaers conceived and designed the experiments, analyzed the data, prepared figures and/or tables, authored or reviewed drafts of the article, and approved the final draft.
- Dirk K. Munsterman conceived and designed the experiments, performed the experiments, analyzed the data, authored or reviewed drafts of the article, and approved the final draft.
- Marcel L. Dijkshoorn performed the experiments, authored or reviewed drafts of the article, and approved the final draft.
- Jeffrey Joël de Groen performed the experiments, authored or reviewed drafts of the article, and approved the final draft.
- Klaas Post conceived and designed the experiments, authored or reviewed drafts of the article, and approved the final draft.

### Data Availability

The raw data are available on MorphoSource:

Morphotype 1A

- NMR-16642, Thoracic vertebra, Pachycetus sp. (small sp.), Sagittal sections, https://doi.org/10.17602/M2/M538333

- NMR-16642, Thoracic vertebra, Pachycetus sp. (small sp.), Axial sections, https://doi.org/10.17602/M2/M538327

- NMR-16642, Thoracic vertebra, Pachycetus sp. (small sp.), Coronal sections, https://doi.org/10.17602/M2/M538321

Morphotype 1B

- NMR-12331, Thoracic vertebra, Pachycetus sp. (large sp.), Sagittal sections, https://doi.org/10.17602/M2/M540213

- NMR-12331, Thoracic vertebra, Pachycetus sp. (large sp.), Axial sections, https://doi.org/10.17602/M2/M540025

- NMR-12331, Thoracic vertebra, Pachycetus sp. (large sp.), Coronal sections, https://doi.org/10.17602/M2/M540016

- NMR-12332, Thoracic vertebra, Pachycetus sp. (large sp.), Sagittal sections, https://doi.org/10.17602/M2/M540234

- NMR-12332, Thoracic vertebra, Pachycetus sp. (large sp.), Axial, https://doi.org/10.17602/M2/M540230
- NMR-12332, Thoracic vertebra, Pachycetus sp. (large sp.), Coronal sections, https://doi.org/10.17602/M2/M540226
- NMR-3404, Lumbar vertebra, Pachycetus sp. (large sp.), Sagittal sections, https://doi.org/10.17602/M2/M540246
- NMR-3404, Lumbar vertebra, Pachycetus sp. (large sp.), Axial sections, https://doi.org/10.17602/M2/M540242
- NMR-3404, Lumbar vertebra, Pachycetus sp. (large sp.), Coronal sections, https://doi.org/10.17602/M2/M540238

Morphotype 2

- NMR-10284, Thoracic or lumbar vertebra, Indeterminable basilosaurid, Sagittal sections, https://doi.org/10.17602/M2/M540789
- NMR-10284, Thoracic or lumbar vertebra, Indeterminable basilosaurid, Axial sections, https://doi.org/10.17602/M2/M540261
- NMR-10284, Thoracic or lumbar vertebra, Indeterminable basilosaurid
Coronal sections, https://doi.org/10.17602/M2/M540256

Morphotype 3

- NMR-10283, Caudal vertebra, Indeterminable basilosaurid, Sagittal sections, https://doi.org/10.17602/M2/M540833
- NMR-10283, Caudal vertebra, Indeterminable basilosaurid, Axial sections, https://doi.org/10.17602/M2/M540822
- NMR-10283, Caudal vertebra, Indeterminable basilosaurid, Coronal sections, https://doi.org/10.17602/M2/M540812

## Supplemental Information

Supplemental information for this article can be found online at http://dx.doi.org/10.7717/peerj.16541#supplemental-information.

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
