# Peer review of "A vertebra of a small species of Pachycetus from the North Sea and its inner structure and vascularity compared with other basilosaurid vertebrae from the same site"

_PeerJ, doi:10.7717/peerj.16541_

## Round 0.1 · original submission · Major Revisions

Dear authors,

I agree with the major issues indicated by the referees, specially concerning several points:

1) the structure of the text (some paragraphs should be moved to their correct sections, e.g. palynological results or biometric methods),

2) the grammatical and typographical errors (please, check carefully all the paper searching for errors before resubmission),

3) excess of figures and tables (include only those that are key for understanding the paper, referee 2 has made suggestions of the most important ones),

4) material description (detail carefully which is the comparison material, and the assignment to particular morphotypes and their description, see referee 2 suggestions at this point), and

5) histology methods and description (see referee 3 comments).

Please, consider the tips and comments of the referees, and provide a new version of the manuscript with changes or an appropriate letter to the referees with your responses.

Thanks in advance,
Blanca Moncunill-Solé

**Language Note:** The Academic Editor has identified that the English language must be improved. PeerJ can provide language editing services - please contact us at [email protected] for pricing (be sure to provide your manuscript number and title). Alternatively, you should make your own arrangements to improve the language quality and provide details in your response letter. – PeerJ Staff

·

Basic reporting

The article is written in unambiguous, clear English text. Paper includes sufficient introduction to the background of the discussed topic. All necessary lliterature referenced in this paper. The article, generally, structured in format of "standart sections". My comment on the structure is about placing the results of palynological analysis to the "Results" section (authors can find this comment in the attached annotated PDF). Figures and tables are relevant and have sufficient resolution. Perhaps there are too much Figures - some of them could be placed in Sipplementary materials.

Experimental design

Method of density measurement should be desribed in the "Materials and metods" section.

Validity of the findings

Authors should focus on uniqueness of NMR-16442 microstructure. Extended comment on this topic can be found in the "Discussion" section in the annotated PDF.

Additional comments

The article contains new valuable data and couclusions about basilosaurid vertebrae microstructure. Vascular system of studied vertebrae described in detailes and with great precision. Authors studied vertebrae of different morphotypes, some of them are unique, so the research has considerable novelty. The paper needs some corrections and additions (all marked in annotated PDF) most of which can be done quickly and easily. I wish success to the authors and wait for the new, improved version of this interesting paper.

Reviewer 2 ·

Basic reporting

This is, first, a manuscript about “a vertebra of a small species of Pachycetus,” but it is also a manuscript about “other basilosaurid vertebrae from the same site,” and about other basilosaurid vertebrae from other sites. As a manuscript, it is long, rambling, and diffuse. The subject should be more focused and the text more tightly written before it is considered for publication. The number of figures is excessive: some are excellent, but most are of little general interest.

Abstract
As a reader I would like to see the “three different large Eocene cetacean taxa” in the abstract documented. Figure 8 purports to distinguish two species. The two scatters in Figure 8 may represent different species, but they may also represent the difference between middle versus posterior thoracics or lumbars within the vertebral column of one species. To illustrate this, I plotted the scatter of length and width for the pachycetine vertebrae of different positions from Morocco (from Gingerich et al., 2022; table S1), on top of the scatters in Figure 8 – which the Morocco vertebrae connect and overlap. Basilosaurid thoracic vertebrae increase greatly in length and width from front to back, so it is essential to know the vertebral positions being compared before attempting any interpretation. The scatter in Figure 8 is not homogeneous, including vertebrae from Het Scheur, but also from North Carolina, Morocco, Germany, and Ukraine. It is not enough to identify these as ‘torso’ vertebrae – the vertebrae change greatly in the torso of one basilosaurid skeleton.

The abstract says “In contrast to the other vertebrae, it was relatively dated by dinoflagellate cysts in adhering sediments. It is one of the very few remains of Pachycetus from Europe, which age could be assessed with reasonable certainty.” OK, this is the abstract, tell us what is the geological age?

The abstract says “the architecture of basilosaurid vertebral centra appears to be basically similar.” OK, this is the abstract, tell us what is this basic similarity?

The abstract says “a remarkable pattern of five interconnected vascular systems is observed.” OK, this is the abstract, tell us what are the five systems observed?

In conclusion, concerning the abstract, it manages to convey a minimum of real information.

Introduction
Kellogg (1936) did not change the name to Platyosphys paulsonii, he placed Zeuglodon paulsonii in the new genus Platyosphys -- which van Vliet et al. (2020) then recognized is a junior synonym of Pachycetus Van Beneden 1883.

Lines 48-49: change “elongated” to “anteroposteriorly elongated” in two places (#5 and #6).

Lines 67-68: Why not show us in one figure all of the 14 vertebrae in the same view at the same scale, say from smallest to largest? This would help the reader understand what are being called three taxa.

Line 71: change “successions” to “strata”.

Materials and Methods
Line 104: Why is NMR-16642 called “Morphotype 1a”? Is this different from 1b – if so how? Here you introduce Morphotype 1, Morphotype 2, Morphotype 3: explain not just what is included, but how these are distinguished? This comes up again on lines 313-325. There is some explanation in Table 2, but this should be included when first raised in the text.

Line 121: Are Padiastrum and Botryococcus genera? Then italicize.

Line 122: Where the authors write “Miocene” I think they mean “Eocene”?

Line 133: Why are “Results of the palynological analysis” buried in the Materials and Methods section of the paper? These are results.

Line 136: Change “presented” to “represented”.

Lines 157-161: Why is information on vertebral measurements buried in a section on “Results of the palynological analysis”?

Results: Systematic Paleontology
Line 213: Why is the material under Pachycetus one posterior thoracic vertebral centrum? There are other Pachycetus vertebrae discussed and illustrated here, and there are some 13 other specimens from the sea floor being sampled.

Line 215: The description should state that this is a description of NMR-16642. The description is good, but there are so few measurements in Table 1 that these should be added at the end of the description paragraph, not orphaned in Table 1 by themselves.

Lines 257-259: It is not surprising that an axial section of NMR-16642 falls completely within the innermost cortical boundaries of NMR-12332 because these seemingly come from different positions within the thoracic series: the first in the middle and the second near the end (judging from development of transverse processes and rib articulations).

Lines 268-270: It is not obvious that the two vertebrae represent different species of Pachycetus.

Line 274ff: I don’t understand the structure of the section headings. Why is there a comparison section under 'Cortex and spongiosa' but not under the other subheadings? And a subheading for 'Comparison' under 'Discussion'? Why are other subheadings not included under 'Discussion'? I ask because a tight organization of these sections and their headings would make it much easier to understand what is included and why it is included.
Inner structure
Cortex and spongiosa—…
Comparison
Microstructure
Cortex—…
Conus—…
Vascularization…
The midvertebral VC—…
The endocortical VC—…
The accessory VC—…
Discussion
Microstructure of bone—…
Vascularisation—…
Comparison—…
Conclusions
Acknowledgements

Line 284: I recommend using ‘cone’ and ‘cones’ in place of Latin conus and coni throughout the text.

Line 301: Why is the Taradell vertebra from Spain included at the end of a paragraph describing compactness of bone in NMR-16642? If included at all, this should be in the following section on Comparison.

Lines 607-627: Conclusions The first paragraph is very clear. In the second paragraph, line 617, what are the earlier observations of Basilosaurus that are contradicted here? Line 490 states that “Vertebrae of Basilosaurus cetoides were not investigated in this study…” so what is the basis for contradicting earlier observations? The third paragraph states that five vascular systems are discerned, but where are these ever defined and described? The word ‘five’ appears four times in the paper, but the five vascular systems are never defined or described.

Tables and table captions
Table 1: I have already stated that Table 1 includes so few measurements that these can be included in the text following line 215. Why is the specimen number sometimes given as NMR 16642 and other times as NMR999100016642? This is confusing for a reader. Explain. Simplify.

Table 2: Here again the specimen numbering system is confusing. The Type 1a and Type 1b separation based on small versus large size is selective and confusing. Are there no vertebral columns with a sequence of associated vertebrae that can be included instead of this selective and seemingly random inclusion of vertebrae from here and there? How can the vertebra in figure 13D be called “shortened”?

Table 3: Tell the reader in the caption what compactness means, what total compactness means, what the surface measures, what the sum value totals, and from what the mean values are calculated. As it stands, this table is incomprehensible.

Table 4: Same questions? Without explanation, this table is also incomprehensible.

Table 5: Where is Table 5?


Figures and figure captions
I would say that figures 1, 2, 4, 5, 6, 7 (without the arrows), and 9 are useful. Figure 15 looks interesting if it can be explained. The rest of the figures should be omitted or moved to in an online supplement.

Figure 1: Balance the labeling on the background map with that on the foreground map. Put the scale on the foreground map, which is the scale relevant for this study.

Figure 2: These colours are bizarre for a geological map and stratigraphic section. Use the standard colours for geological formations and ages (e.g., https://www.geologischekaart.nl/ or Speijer et al. (2020, https://doi.org/10.1016/B978-0-12-824360-2.00028-0).

Figure 3. This chart is incomprehensible. What are the tiny numbers in the left-hand column? Why are the numbered taxa duplicated in two columns? Is this chart necessary – the relevant information is seemingly included in Figure 4.

Figure 4. This chart is helpful, but make the stratigraphic colours match those in revised Figure 2.

Figure 5. This is the principal figure for the paper and shows the vertebra well. Label the scale bar.

Figure 6. Vertebral centra of Pachycetus increase in size from the front to back of the thoracic series, and it isn’t clear these are not both Pachycetus robustus. NMR-16442 is from several positions more anterior in the vertebral column.

Figure 7. Yes, these cross sections are different, but it is very possible that NMR-12332 is from a more posterior position in the vertebral column.

Figure 8. Pachycetus is an unusual whale in that the vertebrae increase greatly in size through the thoracic vertebral column. Thus vertebral position has to be controlled carefully by comparison to a known sequence. It is very difficult to interpret isolated vertebrae, especially when they are not always well preserved.

Figure 9. This is a very interesting figure. Please explain how the right-hand images were “artificially embrightened” [artificially brightened]? Can one simply say here that the darker the bone the greater the density?

Figures 10-23. Yes, it is possible to trace portions of a vascular pattern, but to what end? What is the objectives, the conclusions that emerge? Comparison to human vascularisation seems a long jump physiologically and phylogenetically.

Figure 15. What do the colours mean? Black? Red? Blue? Etc.

Experimental design

No comment. This study is not experimental.

Validity of the findings

No comment.

Additional comments

This manuscript a lot of words and illustrations to describe a 14th example of an isolated archaic whale vertebra dredged from the sea floor offshore on the Belgium-Ntherlands border. Description of one such vertebrae is OK, but pretty meaningless by itself. It is impossible to make any comparative interpretation without explicitly including the previous 13 vertebrae found offshore on the Belgium-Ntherlands border. Also, there are vertebral sequences of closely-related archaic whales described in the literature, and these should be included to bolster the identification of all of the 14 isolated vertebrae to position in the vertebral column and to genus and species.
A much shorter paper could be salvaged from the text here, but to have any real value to science the whole set of 14 dredged vertebrae should be compared in the context of what is known from more complete specimens.

Reviewer 3 ·

Basic reporting

This study reports on a small Pachycetus vertebra. Descriptions of the inner structure and vascularity are presented as obtained through interpretations of CT scans supported by gross anatomy. The paper will form a good new addition to palaeontology. However, at the moment, the manuscript is quite hard to follow because of the confusing expression of the English language. For example, the authors often use phrasing such as “in contrast to the other vertebrae” – which vertebrae?, or “appears to be similar?” – similar to what?, or “has been studied” – this should be “was studied” as it’s past simple tense (the authors use past perfect throughout the manuscript, which is confusing because it implies references to other studies rather than the one conducted by the authors). So, if the authors are able to give the manuscript another review and clarify meaning throughout, it will help the readers tremendously.

The Abstract needs to be re-structured to make it clear what the goal of this study is (i.e. describe the inner structure and vascularity) and what methods were applied prior to concluding the anatomical observations.

In the Introduction, there should also be a clear goal stated. In addition, identifying a gap in the literature about vertebral vascularity/vascularisation processes is needed.

In the Materials and Methods, the paragraph beginning “The relative length of a vertebral centrum is calculated by its dorsal…” appears to be appended to the palynological analysis, but it is a separate section. This should be made clear. Also, here the authors should state that they will undertake a gross anatomical analysis along with the examination of the inner vertebral structure through CT scanning. Here it should be stated which aspects of the vertebral structure will be discussed (vessels, bone organisation, compactness etc.).

Experimental design

I don’t understand the use of the term microstructure at all in this manuscript because it only worked with CT not microCT or histology. The authors are really only looking at inner bone compartments and their structure. The vessel information is not microscopic either, really. The use of the term “microstructure” should be revised. Also, in this vein, the authors might want to clearly state limitations of their study towards the end of the manuscript to note that histology or other microscopic methods could help clarify/validate the presented observations.

The term “vascularisation” actually refers to an active process of gaining vessels. This study does not address this, so “vascularity” is more accurate. Further, the authors should be using anatomically accepted descriptors of vessel orientation (longitudinal, radial and so on) when discussing the vessel results.

Validity of the findings

In Results, on line 278, the authors state “abundant vascular canals” but “abundant” is a vague descriptor particularly that there isn’t another specimen being evaluated/ compared in the same study. Could the authors quantify this? E.g. how many vessels per mm2 approximately.

On lines 281-282, the authors state “Beneath the multi-layered cortex, also spongious-like bone with a more chaotic architecture is present…” – it’s not clear what is less chaotic than this architecture?
In terms of the compactness measurements, Figure 10 illustrates a range of regions or volumes of interest which appear to be randomly selected. Is that the case? Or was there a systematic way of selecting those regions? This needs to be made clear in-text.

In the Discussion, there is a lot of speculation around the vessel morphology and its link to function. I understand the limitations of the literature and the need to include humans as a discussion point, but the authors should make it clear that they’re basically generating new hypotheses rather than providing answers/conclusions. Further, the opening statement “The architecture of the investigated vertebral centra is basically similar to each other” is not clear – what is the similarity?

---

## Round 0.2 · Minor Revisions

Dear authors,

After reading the new version of the manuscript and the comments provided by the referees, I agree with them that some minor points should be changed before publication.

Please, check the manuscript with comments that referees have provided, and the changes that I am also suggesting at the end of this section. Please, take a careful look to the manuscript before re-submitting, to find possible spelling errors.

I would like to congratulate the authors for the large effort that they did in rearranging and restructuring the text, and for all the changes, in this new version of the research. This has improved significantly the quality of the paper.

Minor changes:
Lines 105-106: change “elongated” to “anteroposteriorly elongated” in two places (#5 and #6). Recommendation of referee 2 of the first round.
Line 109-111. “two small species of Pachycetus..” “and Antaecetus”… This is confusing, please explain the recent rename of P. aithai to A. aithai.
Line 113. “Pachycetus” in italics.
Line 133. “et al.” not in italics
Line 143. Delete space before “Fig.”
Line 171-181. I suggest reorganizing the two paragraph. Firstly, that of morphology, with information about the measurements, and following that of bone compactness. This is to follow the same thread as the rest of the manuscript.
Line 186. Include a “)” following “Netherlands”.
Line 206. Delete “(“ before “2020”.
Line 253. Delete “;” following “Subfamily”
Line 317. Delete one “2” in “C22”
Line 414. I think that it should be “microstructure” instead of “inner structure”, because there are two subheading with "inner structure".

Thanks,
Blanca

·

Basic reporting

The article is written in unambiguous, clear English text. Paper includes sufficient introduction to the background of the discussed topic. All necessary lliterature referenced in this paper. The article structured in format of "standard sections". Figures, tables and supplementary materials are relevant and have sufficient resolution. The paper contains significant results and relevant conclusion.

Experimental design

no comment

Validity of the findings

no comment

Additional comments

I have add some minor comments concerning adding compactness values and interpretation of radial vascual canals from previos paper dedicated to basilosaurid vertebra inner structure. I'm sure these minor issuses can be solved quickly and easily. See my comments in attached PDF file.

Reviewer 3 ·

Basic reporting

This is a much stronger manuscript now, the authors did a great job revising the paper.

Experimental design

All clear now to me.

Validity of the findings

New and interesting findings that will contribute to palaeontology.

Additional comments

My one and only very minor comment is to remove the null digits from decimal places when describing some of the results that have no values in decimal places. It will just make it for a cleaner read. So, I literally mean just say 1 mm instead of 1.0 mm etc. I think this could just be done at a proofing stage should the editor accept the article now.

---

## Round 0.3 · accepted · Accept

Dear authors,
Congratulations! In this final version you have addressed all the referees' comments succesfully. From my point of view, this new version has a very clear guiding thread, and all the sections are clear and well explained. When reading it, however, I have noted some minor typographical errors. I detail them below. Please, correct them in the fully typeset publication proofs, together with others that you can identify in them, before final publication.
Cheers,

PhD Blanca Moncunill-Solé


Abstract
Line 9. Change the “,” (comma) following “small species of Pachycetus” and include a “.” (full stop).
Line 20. Delete the comma (,) between “time” and “that”.
Introduction
Parragraph 1, line 13. Add a comma following “Spain”.
Material and methods
‘Morphotype 1a’ paragraph, line 3. Change “NMr-16642” to “NMR-16642”.
Results
‘Palynological analaysis’ paragraph, line 10. The “d” of “dinocyst” in capital letters, “Dinocyst”.
‘Inner structure of the vertebrate used for comparison’ section, “The midvertebrate VC” paragraph. In several places, you have “fossa’s”. Please change to “fossae”.
Discussion
Fist paragraph. Delete “(“ and “)” in “(Houssaye et al., 2015)”.
Conclusions
Last paragraph. Delete “ .” In “the first time are described in detail. . “